# Nintedanib Treatment for Idiopathic Pulmonary Fibrosis Patients Who Have Been Switched from Pirfenidone Therapy: A Retrospective Case Series Study

**DOI:** 10.3390/jcm9020422

**Published:** 2020-02-04

**Authors:** Andrea Vianello, Francesco Salton, Beatrice Molena, Cristian Turato, Maria Laura Graziani, Fausto Braccioni, Valeria Frassani, Dino Sella, Paolo Pretto, Luciana Paladini, Andi Sukthi, Marco Confalonieri

**Affiliations:** 1Respiratory Pathophysiology Unit, Department of Cardiological, Thoracic and Vascular Sciences, University of Padova, 35128 Padova, Italy; beatrice.molena@unipd.it (B.M.); fausto.braccioni@aopd.veneto.it (F.B.); andi.sukthi@aopd.veneto.it (A.S.); 2Pneumology Unit, Department of Medical, Surgical and Health Sciences, University of Trieste, 34149 Trieste, Italy; francesco.salton@asuits.sanita.fvg.it (F.S.); marialaura.gr@libero.it (M.L.G.); marco.confalonieri@asuits.sanita.fvg.it (M.C.); 3Veneto Institute of Oncology IOV–IRCCS, 35128 Padova, Italy; cristianturato@gmail.com; 4Pneumology Unit, Arco General Hospital, APSS Trento, 38062 Arco, Italy; valeria.frassani@apss.tn.it; 5Pneumology Unit, Trento General Hospital, Trento, APSS Trento, 38121 Trento, Italy; dino.sella@apss.tn.it; 6Pulmonology Unit, AS Bolzano, 39100 Bolzano, Italy; paolo.pretto@asbz.it

**Keywords:** idiopathic pulmonary fibrosis, pirfenidone, Nintedanib, adverse event, forced vital capacity

## Abstract

Background: The efficacy and effectiveness of nintedanib as a first-line therapy in idiopathic pulmonary fibrosis (IPF) patients have been demonstrated by clinical trials and real-life studies. Our aim was to examine the safety profile and effectiveness of nintedanib when it is utilized as a second-line treatment in subjects who have discontinued pirfenidone. Methods: The medical charts of 12 patients who were switched from pirfenidone to nintedanib were examined retrospectively. The drug’s safety was defined by the number of adverse events (AEs) that were reported; disease progression was evaluated based on the patient’s vital status and changes in forced vital capacity (FVC) at 12-month follow-up. Results: The numbers of patients experiencing AEs and of the AEs per patient in our study group didn’t significantly differ with respect to a group of 56 individuals who were taking nintedanib as a first-line therapy during the study period (5/12 vs. 22/56; *p* = 0.9999, and 0.00 (0.00–1.00) vs. 0.00 (0.00–3.00); *p* = 0.517, respectively). Two out of the 3 patients who had been switched to nintedanib due to a rapid disease progression showed stabilized FVC values. Conclusions: Nintedanib was found to have an acceptable safety profile in the majority of the IPF patients switched from pirfenidone. Prospective studies are warranted to determine if the drug can effectively delay disease progression in these patients.

## 1. Introduction

A rare disease of unknown etiology, idiopathic pulmonary fibrosis (IPF) is characterized by progressive, irreversible scarring of the lung interstitium leading to a reduction in respiratory function and early mortality [1]. Given that there is no known cure, the goal of IPF treatment is that of stabilizing or even delaying disease progression.

Pirfenidone and nintedanib, the two drugs that have been approved in Europe, the USA, and in some other countries for the treatment of IPF, are characterized by mechanisms of action and pharmacological profiles that differ considerably. Pirfenidone is a small, synthetic molecule drug with anti-fibrotic, anti-inflammatory, and antioxidant properties [2], while nintedanib is a multikinase inhibitor of vascular endothelial, fibroblast, and platelet-derived growth factors [3].

The safety and efficacy of nintedanib as a first-line therapy to reduce the rate of forced vital capacity (FVC) decline and the risk of acute exacerbation [4] have been demonstrated by clinical trials and real-life studies. Data has recently emerged leading to the hypothesis that it could also be an effective second-line treatment for subjects who have discontinued pirfenidone therapy [5]. IPF is known, in fact, to be heterogeneous and patients’ responses to pirfenidone have proven to be quite variable. Some have shown rapid disease progression despite treatment [6]. Pirfenidone has also been linked to considerable adverse events (AEs) leading to discontinuation in approximately 15% of treated patients [7].

This report presents retrospective data examining nintedanib’s safety profile and efficacy in 12 IPF patients who were switched from pirfenidone when they showed signs of drug intolerance or treatment proved ineffective.

## 2. Methods

The medical records of all the IPF patients who were switched to nintedanib after pirfenidone treatment was discontinued during the period between 1 April 2016 and 31 March 2019 and whose clinical and pulmonary function records were complete for at least a 12-month period beginning at the time the patient was switched or who died during that timeframe were collected and reviewed. The data for our study’s analysis were retrieved from the five referral centers composing the “hub-and-spoke network for IPF treatment” of the Triveneto region (Italy). All the study’s participants signed informed consent forms releasing their medical records for review; the data were anonymized. The study is a retrospective case series analysis and has not been prepared according to a research project. For this reason, the need for approval was waived. The study was carried out in accordance with the Declaration of Helsinki of 1975.

IPF was diagnosed at the referral centers in accordance with the multidisciplinary approach recommended by the criteria proposed by the American Thoracic Society, the European Respiratory Society, the Japanese Respiratory Society, and the Latin American Thoracic Association (ATS/ERS/JRS/ALAT) consensus statement [8]. 

At the referral centers, the decision was made to treat with pirfenidone or nintedanib in the first place following a discussion with the patient about the potential side effects.

In the cases in which patients treated with pirfenidone presented serious signs of intolerance despite supportive measures and/or of disease progression, defined as worsening of symptoms and a decline in FVC predicted > 10% over a 12-month period and/or radiological progression [9], the acting physician proposed discontinuing medication and switching to nintedanib.

According to the Italian Medicines Agency (AIFA), nintedanib can be prescribed to patients who are over 40, have a FVC > 50% predicted and a diffusing capacity for carbon monoxide (DLCO) > 30% predicted. Exclusion criteria were: Alanine Transaminase (ALT), Aspartate Transaminase (AST) > 1.5 × ULN, total bilirubin > 1.5 × ULN, high risk of bleeding, INR > 2, PT, PTT > 150% of ULN. Patients who were scheduled for major surgery over the next 3 months’ time or showing a high risk of thrombosis were excluded from treatment. 

Nintedanib was usually prescribed at the recommended full dosage, i.e., twice daily oral doses of 150 mg. Dosage was reduced to twice daily oral doses of 100 mg when AEs presented. When necessary, treatment was temporarily reduced or interrupted altogether, and supportive measures were prescribed. The full dosage regime was reinstated as soon as possible or in the event of persisting, intolerable AEs the acting physician discussed stopping the therapy with the patient.

All the patients were routinely assessed in accordance with the protocol of the referral centers which prescribes a follow-up examination 4 weeks after treatment is prescribed and regular examinations at 3-month intervals thereafter. During each examination patients are questioned about their adherence to treatment and any treatment interruptions or hospitalizations (and their causes) that might have occurred over the preceding three months’ time. An AE checklist, based on the data of clinical trials, was reviewed with each patient at each examination to identify all the signs and symptoms of AEs that might have occurred. Complete pulmonary function testing including DLCO was carried out at the time of each examination. In accordance with the manufacturer’s recommendations, the patient’s liver function was monitored on a monthly basis during the first 3 months after treatment was initiated and at least every 3 months thereafter. A diagnosis of “acute hepatitis” was made according to the presence of abnormal liver function tests accompanied by symptoms of liver injury or jaundice.

The patients’ outcomes, as deduced from their clinical status and pulmonary function parameters at the 12-month follow-up examination, were analyzed. The study’s primary endpoints were the safety and efficacy of nintedanib as a second-line therapy. In accordance with the guidelines of the European Medicines Agency (EMA), the drug’s safety was defined by the number of AEs reported [10]. The number of reported AEs in the study group at the 12 month follow-up examination was compared with that in a group of 56 IPF patients showing similar clinical and pulmonary function characteristics attending our referral centers who were taking nintedanib as a first-line treatment throughout the study period.

The drug’s efficacy was defined by disease progression. Disease progression was evaluated on the basis of the patient’s vital status and change in FVC (median and range); the latter was measured three times: The first time during the period the patient was taking pirfenidone 12 months prior to switching to nintedanib; the second, at the time the switch was made; and the third, 12 months after the patient initiated nintedanib treatment. An absolute decline in FVC predicted >5% within 12 months from nintedanib initiation was consistent with disease progression [11]. Moreover, the number of respiratory-related hospitalizations at the end of the follow-up period was evaluated in comparison with those recorded through the 12-month period prior to switching.

Patient satisfaction with nintedanib was evaluated during follow-up examinations on the basis of a Visual Analog Scale (EQ-VAS; score of 0 to 100), with the endpoints respectively labelled “worst imaginable health state” and “best imaginable health state” [12]; the VAS score calculated at 12 month follow-up examination was compared with that at the time the switch was made. 

The results are expressed, as appropriate, as mean values ± SD, medians, and percentages. The Kolmogorov–Smirnov test was used to check the normality of the data distribution. The continuous variables were compared, depending on the normality of the distributions, using the Student’s *t* test or the Mann–Whitney *U* test. The categorical variables were compared, as appropriate, using the Chi-squared test or Fisher exact test. All statistical calculations were carried out using MedCalc Statistical Software (Ostend, Belgium). A bilateral *p* value < 0.05 was considered statistically significant for all the comparisons.

## 3. Results

### 3.1. The Study Population and Medication Adherence

The data of the twelve patients who were switched to nintedanib were analyzed retrospectively; they represented 11.2% of the total population of patients who were being treated with pirfenidone at the referral centres during the study period (*n* = 107). No patient was switched to pirfenidone after discontinuation of nintedanib during the period. The number of patients who experienced severe AEs leading to treatment discontinuation was similar comparing those treated with pirfenidone to those administered nintedanib (9/107 vs. 4/56; *p* = 0.9999). The reasons pirfedinone was discontinued were disease progression (*n* = 3) and intolerable AEs (*n* = 9), or more specifically, severe photosensitivity reactions (*n* = 6), gastrointestinal disorders (e.g., nausea, dyspepsia, vomiting, diarrhea) (*n* = 2), or elevated liver enzyme values (*n* = 1). The patients’ demographic, clinical, pulmonary, and cardiac function data at the time they were switched to nintedanib therapy are outlined in Table 1. All the patients showed high medication adherence.

### 3.2. Drug’s Safety: Adverse Events

One of the switched patients had intolerable gastrointestinal disorders (i.e., nausea, vomiting, and diarrhea) despite supportive therapy and attempts to reduce drug dosage. It is noteworthy that this same patient reported similar AEs while receiving pirfednidone treatment. One was diagnosed with acute myeloid leukemia. The other seven patients switched from pirfenidone due to intolerable AEs showed better drug tolerability to nintedanib and were able to continue the therapeutic plan. Diarrhea, which was reported by 3 patients, was the most frequent AE. In two cases it led to temporary dose reductions. The third case, which was of a mild intensity, was resolved using concomitant loperamide therapy on a temporary basis. The numbers of AEs per patient and of patients experiencing AEs were essentially the same in the group of patients who were switched from pirfenidone to nintedanib and the group receiving nintedinib as a first line therapy (0 (0–1) vs. 0 (0–3); *p* = 0.517 ; and 5/12 vs. 22/56; *p* = 0.9999, respectively) (Table 2).

### 3.3. Drug’s Efficacy: Clinical Outcomes

At the time of the 12-month follow-up examination, three of the switched patients had died (2 because of congestive heart failure, one had a sudden death). Autopsy examination was not performed in any patient. Two of the switched patients discontinued nintedanid because of intolerable AEs, four showed stabilized FVC values and one showed a decline in FVC (Figure 1). FVC data were unavailable for two of the patients as severe dyspnea impeded them from carrying out pulmonary function testing. Two out of the three patients who had been switched to nintedanib because of rapid disease progression under pirfenidone showed stabilized lung function values. The number of respiratory-related hospitalizations at 12-month follow-up examination was similar to that recorded through the 12-month period prior to switching (0.00 (0.00–1.00) vs. 0.00 (0.00–3.00); *p* = 0.9139). At 12-month follow-up examination, EQ-VAS scores did not significantly differ from those recorded at the time the switch was made (70 (60–90) vs. 70 (50–90); *p* = 0.6455).

## 4. Discussion

Although increasing data on the real-life clinical experiences showing that nintedanib is an effective first-line therapy for IPF patients continue to emerge, only a few studies have investigated its safety and efficacy in patients switched from pirfenidone [5,11]. The data presented here suggest that nintedanib is a well-tolerated second-line therapy whose side effects are similar to those found in patients receiving it as a first-choice treatment. It is important to remember that the patients in our study group may have had a propensity towards AEs given that most of them had proven to be intolerant to pirfenidone. Our results confirm in any case that there are no important safety issues that are linked to the use of nintedanib in routine clinical settings.

In accordance with the findings of clinical trials, the most frequent AE in our case series was diarrhea, but the percentage of patients (25%) presenting the condition was lower with respect to that reported by other real-life studies (between 50% and 63%) [13,14,15]. In line with our results, another retrospective study examining the records of 64 IPF patients, nearly half of whom originally treated with pirfenidone, showed that diarrhea, which was generally considered manageable, was reported by only 33% of the patients [11]. The low rate (8.3% = 1 out of 12 cases) of permanent treatment discontinuations linked to gastrointestinal disorders found in our patients was moreover comparable to the data of real-life studies investigating nintedanib as a first-line therapy [11,16]. 

The small number of patients studied here prevents us from drawing any conclusions on the comparative effect of the two drugs in attenuating disease progression, as reflected in the FVC decline rate. We can nevertheless report that switching to nintedanib was associated with FVC stabilization in 2 (out of the 3 patients) who had shown accelerated disease progression while they were taking pirfenidone. The metabolic heterogeneity of IPF involving nine different pathways in human lungs [17] could hypothetically explain why both the intra and inter-individual responses to the two agents were so different in our study population.

Even though nintedanib treatment has been associated with a lower risk of IPF acute exacerbation [18], in our experience the frequency of respiratory-related hospitalizations did not significantly reduce during the follow-up period after switching.

Finally, it should be noted that patient satisfaction with nintedanib was high (EQ-VAS score: 70 (60–90)), suggesting that this drug was well tolerated as a second-line therapy, despite the relevant proportion of patients (over 40%) who experienced AEs.

The study presents some limitations: First and foremost, and as noted above, the small number of patients enrolled and studied. Secondly, analyzing the data of only the patients with complete FVC records (and thus able to take pulmonary function tests) could have biased towards those patients with less severe disease progression. One of the study’s major strengths was that the diagnosis of IPF was provided by expert physicians working at specialized clinics.

Despite these limitations, an analysis of the data collected has important implications, we believe, for clinical care, suggesting that nintedanib can be considered a safe option for IPF patients proving to be intolerant to pirfenidone. Larger prospective studies are warranted to determine if the drug is effective in delaying disease progression in subjects showing rapid progression while they are receiving pirfenidone treatment.

## 5. Conclusions

In conclusion, nintedanib was found to have an acceptable safety profile in the majority of the IPF patients switched from pirfenidone. 

Due to the limited number of cases, we were prevented from determining if its use as a second-line therapy can effectively decrease the rate of disease progression.

## Figures and Tables

**Figure 1 jcm-09-00422-f001:**
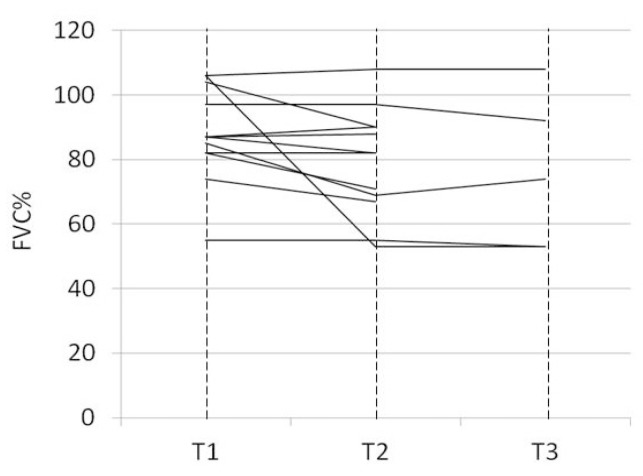
Change in forced vital capacity (FVC)% predicted under pirfenidone and nintedanib treatment. Each line represents one individual patient. T1 = Time-point 1, under ongoing pirfenidone treatment 12-months prior to switch to nintedanib; T2 = Time-point 2, 12-months later at the time the switch was made; T3 = time-point 3, 12-months after the patient initiated nintedanib treatment.

**Table 1 jcm-09-00422-t001:** Patients’ demographic, clinical, and pulmonary-cardiac function characteristics at the time of switching to nintedanib. (BMI = body mass index; DLCO = diffusing capacity for carbon monoxide; FVC = forced vital capacity; FEV_1_ = forced expiratory volume in the 1^st^ second; NIV = non-invasive ventilation; PH = pulmonary hypertension).

**Age (years), Median (range)**	76 (31–79)
**Gender (males/females)**	10/2
**BMI (kg/m^2^), Median (range)**	25.71 (20.20–33.67)
**Number. of previous smokers**	11 (84.6%)
**Number of patients who underwent lung biopsy**	1 (8.3%)
**Length of time from diagnosis to treatment initiation (months), Median (range)**	16 (160)
**Number of hospitalizations earlier in the year median (range)**	0
**Number of co-morbidities, median (range)**	1 (0–4)
**Number of patients with previous cardiac disease**	5 (41.6%)
**Number of patients with PH**	4 (33.3%)
**Number of patients previously administered steroids**	8 (66.6%)
**Number of patients previously administered other immunosuppressive therapy**	4 (33.3%)
**Number of pts administered long-term oxygen therapy**	7 (58.3%)
**FVC, L median (range)**	2.06 (1.53–3.15)
**FVC, % median (range)**	79.5 (53.0–106.0)
**DLCO**, **ml/min/mmHg, median****(range)**	7.02 (5.14–24.37)
**DLCO, % median (range)**	31.0 (30.0–51.0)

**Table 2 jcm-09-00422-t002:** Adverse events reported over the period of observation in patients switched to nintedanib and those who received nintedanib as a first-line therapy. (AE = adverse event; ALT = alanine transaminase; AST = aspartate transaminase; Pts = patients; ULN = upper limit of normal).

	Patients Switched to Nintedanib (*n* = 12)	Patients Who Received Nintedanib as a First-Line Therapy (*n* = 56)	*p*-Value
Age (years), median (range)	76 (31–79)	75(55–88)	0.8637
Gender (males/females)	10/2	42/14	0.7171
BMI (kg/m^2^), median (range)	25.71 (20.20–33.67)	26.67 (14.00–37.10)	0.1605
Number of previous smokers	11 (84.6%)	35 (62.5%)	0.1703
Length of time from diagnosis to nintedanib initiation (months), median (range)	16 (1–160)	2(0–83)	0.0001
Number of co-morbidities, median (range)	1 (0–4)	1 (0–5)	0.893
Number of gastrointestinal AEs Diarrhea Anorexia, weight loss	3 (25%) 0 (0%)	22 (39.3%) 4 (7.1%)	0.5130 0.9999
Number of other AEs AST/ALT above the ULN Acute hepatitis Myalgia Acute myeloid leukemia	1 (8.3%) 0 (0%) 0 (0%) 1 (8.3%)	7 (12.5%) 1 (1.8%) 1 (1.8%) 0 (0%)	0.9999 0.9999 0.9999 0.9999
Number of AEs per patient, median (range)	0 (0–1)	0 (0–3)	0.517
Number of patients experiencing AEs	5 (41.7%)	22 (39.3%)	0.9999
Number of patients who required temporary treatment discontinuation or dose reduction	4 (33.3%)	18 (32.1%)	0.9999
Number of patients who discontinued treatment due to AEs	2 (16.6%)	4 (7.1%)	0.2846

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
