# Peer review of "Nintedanib Treatment for Idiopathic Pulmonary Fibrosis Patients Who Have Been Switched from Pirfenidone Therapy: A Retrospective Case Series Study"

_jcm, 2020, doi:10.3390/jcm9020422_

Round 1

Reviewer 1 Report

The article entitled “Nintedanib treatment for idiopathic pulmonary fibrosis patients who have been switched from pirfenidone therapy. A retrospective case series study” describes a total of 12 patients who are switched from pirfenidone to nintedanib.

Major;

Though the efficacy and tolerability of nintedanib after pirfenidone is of scientific interest, the result is confined to a small number of patients, and the number of patients who could show the change in forced vital capacity (FVC) is limited to only 5 patients.

In addition, the clinical outcome is not clear which should be defined in the Methods section, and the clinical benefit obtained from nintedanib in the second line settings should be mentioned in the Result section.

In the Results section, the adverse events are mainly described. As to the change in FVC, the authors particularly emphasized that 2 out of 3 patients were stabilized with nintedanib treatment who had shown rapid disease progression under pirfenidone. Under the circumstances with a small number of patients with objective changes, subjective improvement assessed by patient reported outcomes could be useful.

On the other hand, the data include as many as 107 patients with first line treatment with pirfenidone and 56 patients with nintedanib. Therefore, the authors could revise the manuscript to include 107 patients with pirfenidone in the first line setting and show the difference between pirfenidone and nintedanib, which could be helpful for readers.

Minor;

i) The word "adverse events" is sometimes mistaken as "adverse effects" in line 71 and 161, or "side effects" in line 190. 

ii) JRS (Japanese Respiratory Society) is lacking in the consensus statement (line 90).

Reviewer 2 Report

 Andrea Vaniello and colleagues have produced an interesting and succinctly written manuscript.  The topic is of interest given the significant issues with side effects reported in both the antifibrotic clinical trials and real-world data.  As the authors mention, there is little published data describing patients who have switched between antifibrotics. 

The study is however significantly limited by the small number of patients included, which makes it difficult to draw firm conclusions from the data presented. 

There are several areas that need attention.

Major changes

Methods, Line 129-130. The authors define disease progression as a decline in FVC of >5%.  Is this absolute or relative decline?  Generally, a 10% decline in FVC is more consistently used in the literature as this is significantly predictive of mortality.  Is there a reason why 5% was used?  Regardless, in the subsequent analysis, there is no explicit mention of the number of patients who had a >5% decline.  Could this data be added more explicitly? Results, table 2. Could the number of patients who discontinued treatment due to AEs be recorded in the table (and compared between groups).  This is mentioned in part in line 176-177.  Could the number of patients who required temporary discontinuation or dose reduction also be included? Conclusions – this paragraph is not a conclusion but should be included in the discussion. The conclusion should be a brief summary of he findings described in the manuscript.

Minor changes

Introduction, Line 60. I don’t think the trade names (Esbriet and Ofev) are required. Introduction, Line 67. I think “But some” can be removed to avoid the sentence starting with “But” Introduction, Line 69-70. I would place an “and” between “heterogeonous” and “patients” to improve sentence structure.  I would remove “and” from “and some have..” and start a new sentence. Methods, Line 82-83. I don’t think the sentence “Instituted by …. expand IPF treatment” adds anything and should be removed. Methods, line 131. Could the authors add “medians and range” Results, table 1. Length of time to diagnosis – the range has 2 decimal places but the median has 0.  Could this be consistent.  0 decimal places would be more appropriate.  Likewise Hospitalizations could have 0 decimal places.   FVC % and DLCO % could have 1 decimal place. Results line 157. Change “vomit” to “vomiting”. Results line 165. I would change “substantially” to “essentially” Results line 166-167. I would change the median and range to 0 or 1 decimal place. Results, table 2. Could “Acute hepatitis” be defined in the methods. Results, table 2. Could decimal places be adjusted as described above. Discusion line 190. I think the term “demonstrate” should be changed to “suggest” to temper the conclusions drawn from the data, given the limitations of the study.

Round 2

Reviewer 1 Report

The manuscript has been revised in accordance with the comments for the authors.

It is good to show efficacy of nintedanib on subjective symptom using EQ-VAS scores.

However, it is too strong to conclude that nintedanib as a 2nd line therapy is associated with reduction in the rate of disease progression. The author should recognize that only 2 out of 3 patients with rapid decline in FVC were temporarily stabilized. Since IPF is the most dominant idiopathic interstitial pneumonia, it is not rare and the result with a limited number of patients cannot be generalized.

Thus, the 2nd sentence in Conclusion part is not appropriate.